# Effects of Dry–Wet Cycling and Temperature on Shear Strength and Microscopic Parameters of Coal Measure Soil

**Gang Huang** [1,*] and **Mingxin Zheng** [2]

1   School of Civil Engineering, Huanggang Normal University, Huanggang 438000, China
2   School of Transportation Engineering, East China Jiaotong University, Nanchang 330000, China
*   Correspondence: huanggang0317@163.com; Tel.: +86-134-0990-2162

**Abstract:** Exposed coal measure soil (CMS) found in the mountains of Southern China is significantly affected by the seasonal climate, which makes this region prone to frequent shallow landslides. In this regard, very few studies have focused on the shear strength and microscopic characteristics of CMS subjected to dry–wet cycling and temperature. The aim of this study was to experimentally investigate the effects of dry–wet cycling and temperature on shear strength and microscopic parameters of CMS. We carried out an unconsolidated undrained triaxial test and scanning electron microscopy of CMS obtained from the K209 slope on the Chang-li highway. Our results indicated that the soil shear strength and microstructure parameters significantly decreased before three dry–wet cycles. Above 35 °C, the temperature affected mainly the mean fractal dimension. The soil cohesion was negatively correlated with the fractal dimension and positively correlated with the probability entropy. The surface-crack occurred once the stress value of high temperature was greater than 0.57 MPa. Strain-softening, swelling–shrinkage, low soil strength, and high soil temperature formed the main factors underlying rainfall-induced K209 shallow landslides.

**Keywords:** coal measure soil; dry–wet cycles; shear strength; temperature; shallow landslides

## 1. Introduction

Coal measure soil (CMS) is a new type of soil encountered in slope engineering wherein the coal-measure strata outcrop and undergo weathering. CMS is primarily distributed in the hilly and mountainous areas of South China. In recent years, the coal-measure soil (C-M-S) area has witnessed an increasing number of shallow landslides (generally less than 2 m deep), and the failure surface is parallel to the slope surface. 30–50% of the landslides had reoccurred even after treatment. As a matter of concern, frequent landslides in the C-M-S area have resulted in substantial economic losses and sometimes even directly endangered human lives. Moreover, during the construction and operation of highways in such regions, a major challenge involves controlling the stability of the coal-measure-soil (C-M-S) slope and preventing the occurrence of landslides or soil erosion. When compared with other rock-soil slopes, CMS exhibits many complex properties, such as discontinuity, nonlinear constitutive behavior, anisotropy, and heterogeneity [1,2]. The C-M-S structure is such that it weathers and degrades easily when subjected to seasonal dry–wet cycles, and the resulting situation satisfies the conditions for crack generation within the soil mass. Moreover, this configuration is more prone to landslides under rainfall. Therefore, a close examination of the mechanical properties and failure mechanisms under dry–wet cycling is essential to assess and control the stability of C-M-S slopes.

Many researchers have studied the physical and mechanical characteristics of CMS. Hu [3] applied the direct shear test to demonstrate that the shear strength of CMS exhibits an obvious correlation with the initial moisture content. Here, we note that water and soil interactions are the result of a combination of the soil macrostructure and microstructure. In this regard, Zuo [4] acquired a C-M-S sample from the Chenzhou landslide to qualitatively

expound the inducing factors and deformation mechanism. Along these research lines, the discrete element method (DEM) is used to perform 3D numerical simulations of the triaxial shear test for CMS [5]. Zhang [6] analyzed the effects of the particle shape and size on the physical and mechanical characteristics of soil mass via numerical triaxial-test simulations. Liao [7] researched the variation pattern and stress distribution characteristics of moisture content in rainfall for C-M-S slopes. Here, we note that all these studies focus on the C-M-S physical and mechanical properties of CMS under different water contents. However, very few studies have examined the C-M-S strength behavior under dry–wet cycling.

Notably, dry–wet cycling is a familiar shallow-soil weathering degree determined by short-term rainfall and sun exposure. The soil strength characteristics and microstructural features affected by dry–wet cycling have received widespread attention. Moreover, surface-crack occurrence and development in rock soil are generally related to dry–wet cycling [8,9]. Furthermore, dry–wet cycling is a crucial reason underlying the deterioration of microstructure and macro mechanics of a wide range of unsaturated soils, such as expansive soil [10,11], loess [12], and red clay [13,14]. In recent years, researchers have also studied the C-M-S strength characteristics subjected to dry–wet cycling [15,16]. In this regard, Yang [17] studied the variability in the soil–water characteristic curve (SWCC) of CMS under dry–wet cycling. Maio [18] proved that the suction had a positive effect on the shear strength. Fredlund [19] analyzed the adsorption strength and shear strength between soil particles affected by matric suction. Here, we note that C-M-S research has mainly been confined to study the effects of dry–wet cycling on soil strength at normal temperature. However, most previous studies have not accounted for summer temperatures that can generate enhanced cracks in rock soil. Moreover, the combined effects of different temperatures and dry–wet cycling on the C-M-S strength have been ignored.

As regards general studies on the effects of temperature on soil strength, some theories and methods have been proposed. Mitchell carried out triaxial strength tests on saturated silt in the temperature range 0–35 °C and found that the soil strength decreases with increasing temperature [20]. Moreover, high temperature has been reported to reduce the soil strength and enhance crack formation [21]. Cabalar found that higher temperature leads to a jump in pore water pressure [22]. We determined that the strength of the red clay and expansive soil decrease with increasing temperature (from −4 to 60 °C), and red clay, in particular, exhibits higher heat sensitivity [23]. Moreover, Roshani described the effect of temperature on the SWCC under isothermal environment [24]. However, the above-mentioned studies focus on the influence of temperature on the strength of clay, silt, red clay, and expansive soil. Studies on the effect of temperature on the C-M-S strength are still scarce.

Meanwhile, extensive studies have indicated that changes in soil macroscopic mechanics are generally related to the microstructural features [25,26]. Furthermore, water is a vital cause of mechanical deterioration in unsaturated soils [27]. In this regard, scanning electron microscopy (SEM) has become the most commonly used geotechnical technology in the study of geotechnical microstructure. The microstructure and morphological characteristics of CMS along the shear plane for different water contents are studied via SEM and direct shear tests [28]. CMS contains a large number of clay minerals, which have expansive characteristics [29]. In addition, surface soil degradation caused by seasonal dry–wet cycles destroys the surface structures used for slope protection. Thus, it is important to understand the soil-failure mechanism via analyzing the C-M-S microstructures and the variation in the soil shear-strength parameters under dry–wet cycling. Moreover, the coupling effects of dry–wet cycling and temperature on the microstructural features of CMS have not, thus far, been investigated in detail.

This paper firstly performs surveys on the shear strength and microscopic parameters of coal measure soil under dry-wet condition and seasonal climate in the mountains of Southern China. In order to further explore the effects of dry–wet cycling and temperature on the shear strength and microscopic parameters of CMS, an unconsolidated undrained triaxial test and SEM test were carried out. Firstly, the effects of dry–wet cycling and

temperature on the shear strength of C-M-S samples were investigated. Then, the microscopic parameter changes in the CMS after dry–wet cycling was studied. Finally, the crack development relationship with the temperature-induced stress after dry–wet cycling and the failure mechanism of the K209 landslide on the Chang-li highway were discussed.

## 2. Materials and Methods

### 2.1. Engineering Geological Properties

The geographical region considered in the study is located to the west of Jiangxi Province, which is a subtropical zone. The region's terrain consists of an interphase arrangement of horst fault blocks, graben hills, and valley basins. The overlying layer of CMS is formed by the Permian Longtan Formation ($P_2l$) and Triassic Anyuan Formation ($T_3a$).

Climatically, the weather changes in the study area are diverse. The annual average temperature is 17–21 °C. The extreme maximum surface temperature is 61.6 °C, and the lowest surface temperature is −8.8 °C [30]. Since 2013, many shallow landslides have happened along the Chang-li highway (Figure 1). Furthermore, as shown in Figure 2, the engineering classification of CMS can be estimated as powder sand soil SW according to the engineering classification standards [31]. The physical properties of the CMS are listed in Table 1. Moreover, this study considers the K209 landslide as an example to understand the mechanism of C-M-S landslide.

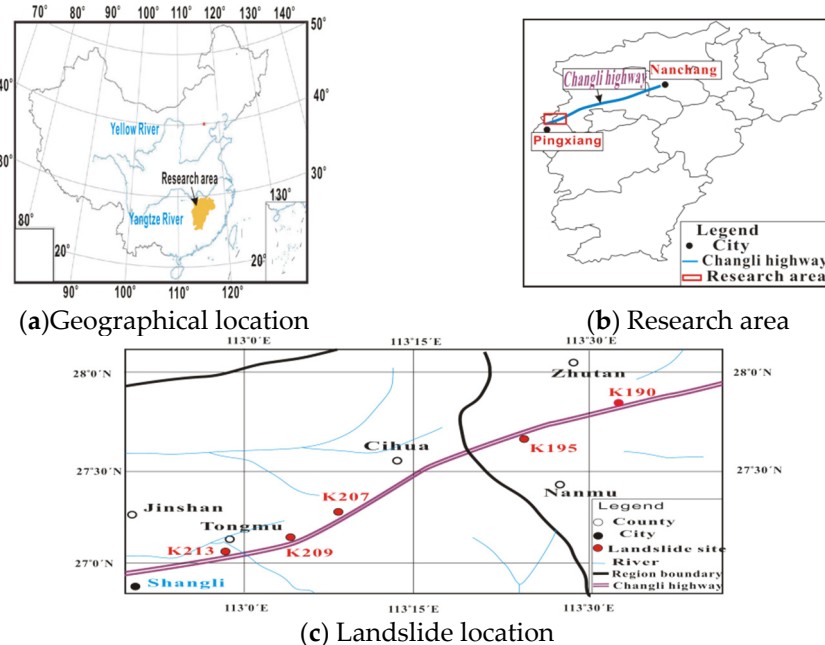

(**a**)Geographical location  (**b**) Research area

(**c**) Landslide location

**Figure 1.** Location of the study area and landslide grouping.

**Table 1.** C-M-S physical properties.

| Landslide Sites | Dry Density/kg·m$^{-3}$ | Optimal Moisture Content/% | Liquid Limit/% | Plastic Limit/% |
|---|---|---|---|---|
| K213 | 1.65 | 10.21 | 42.10 | 31.10 |
| K209 | 1.70 | 9.54 | 39.30 | 27.50 |
| K207 | 1.59 | 11.58 | 45.80 | 34.10 |
| K195 | 1.67 | 10.12 | 47.50 | 33.40 |
| K190 | 1.75 | 9.78 | 48.00 | 34.60 |

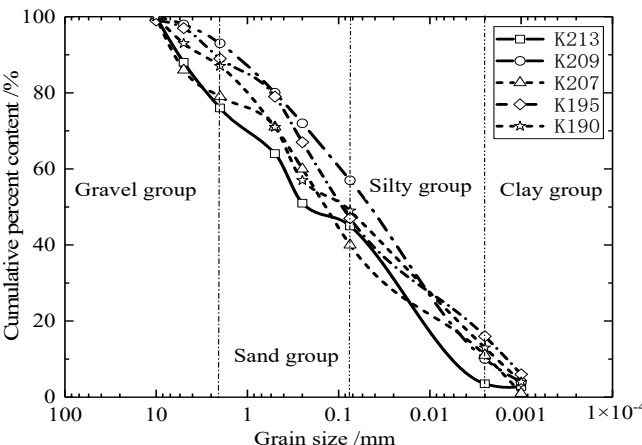

**Figure 2.** Grain-size distribution of the C-M-S samples.

### 2.2. Shear-Strength Testing Experimental Procedure

The CMS collected from K209 was placed in a plastic bag, sealed for storage, and placed in an oven operating at 70 °C for 72 h. These preparation procedures of dried samples were crushed, passed through a 2-mm sieve, adjusted to an optimal moisture content of 9.54%, and fashioned into a standard triaxial specimen (39.1 mm diameter and 80 mm height), as per the standards applicable to Chinese standard GBT50123-2019 [32].

Triaxial tests were carried out after the C-M-S specimens were subjected to 5 dry–wet cycles, which included a wetting step and a drying step. The development characteristics of soil fissures were affected by the water-content variation. The dry–wet cycle under atmospheric conditions was mainly caused by rainwater infiltration, rising groundwater level and transpiration, which was similar to 1D moisture transfer. Polyvinyl chloride wrapped on the laterally sides. We injected 10% of water by weight on the upper and lower sides of the sample and stored water for 4 h. Then, the samples wrapped by Polyvinyl chloride and rubber film were dried at 70°C for 8 h in a climate chamber. The total period of each dry–wet cycle was 12 h. The preparation procedure for the C-M-S specimens is shown in Figure 3.

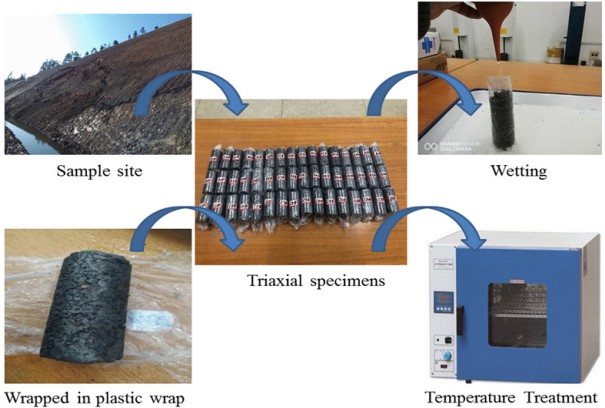

**Figure 3.** Preparation procedure of C-M-S specimens.

The specimens were drawn in a vacuum and saturated for 4 h via continuous pumping. The bulk density of the saturated specimen was estimated as 1.57 g/cm$^3$. The specimen-confining pressure was set to a constant loading rate of 0.2 kPa/min. Next, axial pressure was applied to the upper and lower surfaces of the soil specimen at a constant axial strain rate of 0.05 mm/min. Triaxial tests were performed on the specimens under three confining pressures: 100, 200, and 300 kPa.

The hydraulic behavior of unsaturated soils could be characterized by the soil-water characteristic curves (SWCCs) under wetting and drying cycles. We performed the pressure plate device method (axis translation technique) to obtain SWCC. The SWCC and van Genuchten (1980) equation fit of the C-M-S specimens are shown in Figure 4. It was noted that the air entry value of no cycle and 5 cycles is about 20 kPa and 11 kPa, respectively. The dry–wet cycling resulted in the densification of C-M-S unsaturated soils, which led to changes in SWCC.

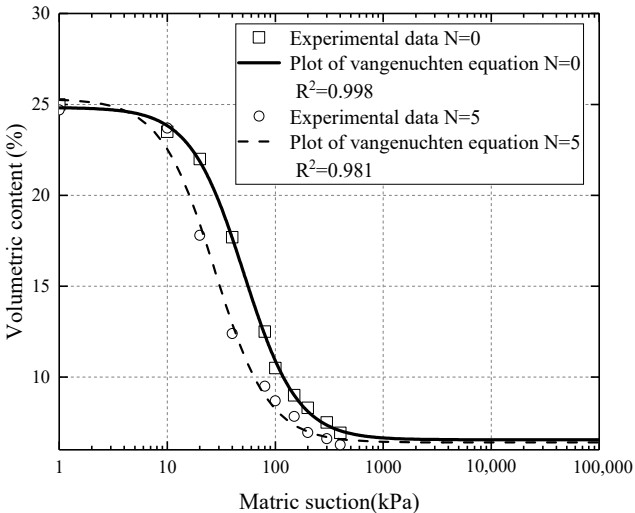

**Figure 4.** The SWCC of the C-M-S specimen.

Based on the climate of the study area, the temperature change experiment was carried out under the condition of shear strength. The temperature condition of the specimen was controlled by heating or cooling water in the triaxial pressure chamber. The heating sheet was used to control the heating water, and the cooling sheet was used to control the cooling water during the test. We set three different water temperatures within the pressure chamber: 0 °C, 35 °C, and 70 °C. Temperature-controlled water was circulated from an external tank to the pressure chamber. Water at three different water temperatures was pumped to the pressure chamber by an electric pump and returned to the external tank by a siphon system. Special cooling equipment and hot plates were used to control the water temperature in the external water tank.

*2.3. Quantitative Analyses of Soil-Sample Microstructure*

To quantitative examine the effects of dry–wet cycling and temperature on the C-M-S microstructure, we obtained the SEM images of the reserved specimens and processed them with IPP 6.0 software (Figure 5a). Here, we noted that the microfracture transfixion probability was bound up with the pore structure of the soil [33]. Thus, we considered the two microscopic parameters of the plane fractal dimension and probability entropy. These two microscopic parameters reflected the pore shape and particle arrangement characteristics of the C-M-S specimens [34]. As described in Table 2, the dominant chemical compositions of the C-M-S specimen were $SiO_2$, $K_2O$, $CaO$, and $Al_2O_3$.

**Table 2.** Chemical composition of the C-M-S specimen.

| Specimen | $SiO_2$ | $K_2O$ | $CaO$ | $Al_2O_3$ | $F_2O_3$ | $MgO$ | $Na_2O$ | Loss on Ignition |
|---|---|---|---|---|---|---|---|---|
| CMS | 32.54 | 17.24 | 15.20 | 14.58 | 2.24 | 1.40 | 0.75 | 16.05 |

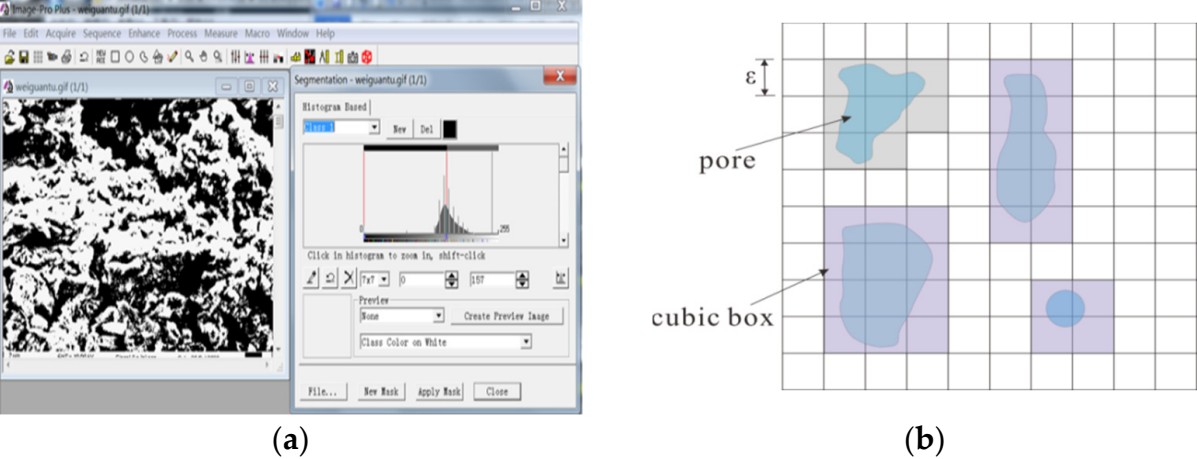

**Figure 5.** (**a**) IPP 6.0 software analysis, (**b**) Box-counting dimension diagram.

Here, we noted that the irregular and non-smooth geometry in nonlinear systems have been widely used in fractal theory [25]. The degree of aggregation in soil mechanics was lower when the plane fractal dimension was larger. The purpose of this study was considered the plane fractal dimension $D_\rho$ to study the C-M-S microstructure fractal and estimated this parameter via box-counting (Figure 5b). The plane fractal dimension of the soil particle was a geometric object that indicates the aggregation of a soil in the plane space. The value of plane fractal dimension was obtained by the ratio of the cubic box length occupied by the total grid number occupied by the target, as per the following formula.

$$D_\rho = -\lim_{\varepsilon \to 0} \frac{\lg N(\varepsilon)}{\lg(\varepsilon)} \tag{1}$$

where $\varepsilon$ is the cubic box length, $N(\varepsilon)$ is the total grid number occupied by the target.

Probability entropy $H_n$ could be used to determine the order of the C-M-S pore unit. A larger $H_n$ value implied a greater degree of pore disorder in the CMS. This parameter could be expressed as

$$H_n = -\sum_{i=1}^{n} F_i(\alpha) \log_n F_i(\alpha) \tag{2}$$

where $F_i(\alpha)$ denotes the incidence of pores in each directional angle interval $\alpha$. Moreover, $F_i(\alpha) = \frac{n_\alpha}{n}$, $n$ denotes the total number of pore units and $n_\alpha$ the number of pore units lying along the orientation angle in the interval $[0, \pi]$. The $H_n$ value can range from 0 to 1.

## 3. Test Result Analysis

### 3.1. Stress–Strain Curve

Upon conducting unconsolidated consolidated undrained (UU) triaxial tests, we obtained the stress–strain curves of Fifty-four C-M-S specimens under different dry–wet cycles and temperatures. The stress–strain curves under the three confining pressures are plotted in Figure 6. The shear strength was considered as corresponding to the peak deviator stress or the deviator stress corresponding to 15% axial strain (when there was no peak) of the stress–strain curve.

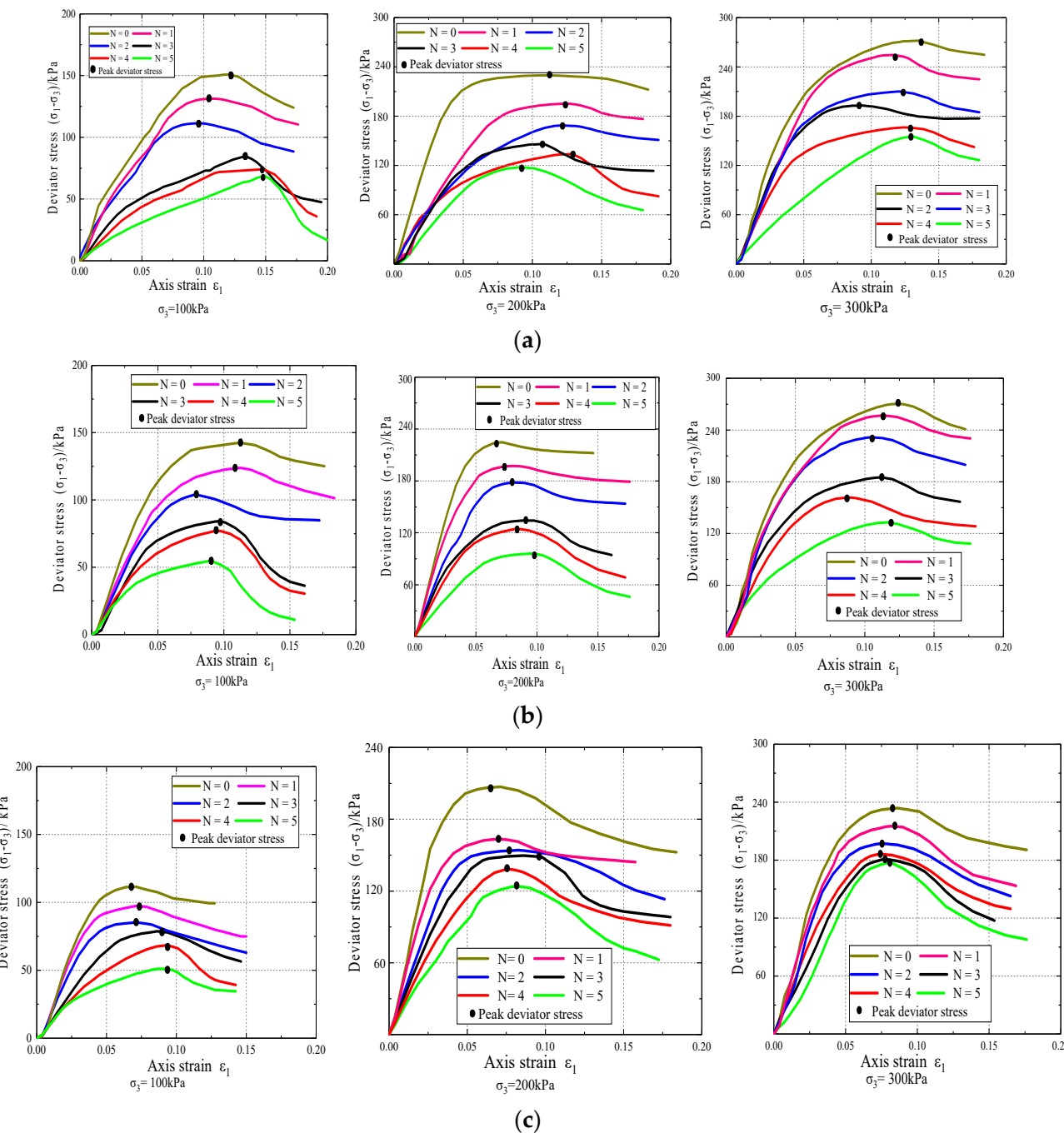

**Figure 6.** Stress–strain curves of C-M-S samples. (**a**) 0 °C, (**b**) 35 °C, (**c**) 70 °C.

From Figure 6, we noted that the stress–strain curves exhibit a two-stage nonlinear relationship: (a) Elastic–plastic stage: the stress–strain curve was nonlinear below the peak deviator stress. (b) Strain softening stage: the deviator stress decreased with increasing strain after the CMS reaches its peak deviator stress.

The shear strength index (cohesion $c$ and net friction angle $\phi'$) and microscopic parameters (plane fractal dimension $D_\rho$ and probability entropy $H_n$) under different dry–wet numbers ($N$) and temperatures ($T$) are shown in Table 3. The peak deviator stress of the CMS under different confining pressures was lower than 277.10 kPa. This result indicated that the shear strength of the CMS was very low. Moreover, a higher temperature corresponded to a lower peak deviator stress. For instance, at the confining pressure of 100 kPa and 0 cycles, the deviator stress at 0 °C was 150.79 kPa, which was 1.34 times the stress at

70 °C (112.49 kPa). This result indicated that the temperature significantly degrades and deforms the CMS. In addition, for a given confining pressure, the peak deviator stress of the CMS sample decreased gradually with increasing temperature.

**Table 3.** The parameters of peak deviator stress and shear strength.

| $T$ | $N$ | Peak Deviator Stress/kPa | | | $c$ | $\phi'/°$ | $D_\rho$ | $H_n$ |
|---|---|---|---|---|---|---|---|---|
| | | 100 kPa | 200 kPa | 300 kPa | | | | |
| 0 °C | 0 | 150.79 | 234.94 | 277.10 | 36.50 | 14.00 | 1.324 | 0.299 |
| | 1 | 131.01 | 190.77 | 255.16 | 26.80 | 13.71 | 1.330 | 0.275 |
| | 2 | 101.67 | 168.07 | 225.20 | 16.30 | 13.65 | 1.336 | 0.242 |
| | 3 | 87.62 | 150.85 | 203.04 | 12.60 | 12.94 | 1.344 | 0.232 |
| | 4 | 74.51 | 135.06 | 175.21 | 11.10 | 11.65 | 1.350 | 0.227 |
| | 5 | 69.24 | 129.66 | 163.75 | 10.70 | 11.07 | 1.351 | 0.225 |
| 35 °C | 0 | 143.18 | 230.36 | 266.23 | 34.70 | 13.77 | 1.325 | 0.293 |
| | 1 | 123.38 | 181.24 | 245.93 | 24.00 | 13.59 | 1.327 | 0.27 |
| | 2 | 104.27 | 167.57 | 231.08 | 16.00 | 13.95 | 1.333 | 0.236 |
| | 3 | 80.12 | 136.39 | 189.20 | 11.50 | 13.71 | 1.342 | 0.222 |
| | 4 | 72.22 | 126.41 | 170.23 | 10.20 | 11.36 | 1.352 | 0.214 |
| | 5 | 52.32 | 99.81 | 140.21 | 9.40 | 10.60 | 1.354 | 0.208 |
| 70 °C | 0 | 112.49 | 225.10 | 254.87 | 30.30 | 13.89 | 1.328 | 0.289 |
| | 1 | 97.37 | 176.34 | 232.32 | 21.30 | 13.36 | 1.335 | 0.262 |
| | 2 | 87.37 | 153.77 | 212.45 | 15.70 | 12.89 | 1.338 | 0.231 |
| | 3 | 78.08 | 148.76 | 207.23 | 11.00 | 13.36 | 1.347 | 0.221 |
| | 4 | 70.13 | 132.56 | 186.99 | 9.50 | 12.36 | 1.356 | 0.215 |
| | 5 | 50. 31 | 129.02 | 171.29 | 9.00 | 11.65 | 1.359 | 0.205 |

*3.2. Effect of Dry-Wet Cycling and Temperature on Shear-Strength Parameters*

To further explore the effects of dry–wet cycling and temperature on the shear strength parameters, we plotted the curves of the cohesion and friction angle values obtained from all tests for the C-M-S specimens in Figures 7 and 8. From Figure 7 we noted that the C-M-S cohesion after dry–wet cycling was less than that of the intact CMS. The C-M-S cohesion decreased with increasing in the $N$ and stabilizes after a certain $N$. At 35 °C, the soil cohesion decreased from 34.7 to 9.4 kPa as the $N$ increased from 0 to 5, corresponding to a 61.94% decline in cohesion over 5 cycles. This result revealed that the C-M-S cohesion was strongly affected in the first three cycles. As can be observed in Figure 7, the cohesion decreased with increasing temperature. The cohesion after the third cycle decreased from 12.6 to 11.0 kPa as the temperature was raised from 0 to 70 °C. It was also noteworthy that the cohesion decreased by a greater degree at higher temperatures after the second cycle. This was because each dry–wet cycle destroyed the soil structure, the temperature limited stress drop, and the cohesion reduced faster [35]. The observed linear-reduction trend in the cohesion could be expressed by the following formula:

$$c_{\mathrm{T}} = c_0 + a_1 (T - T_0) \tag{3}$$

where $T$ and $T_0$ denote the temperature and initial temperature, respectively, $c_T$ and $c_0$ denote the cohesion for temperature $T$ and initial cohesion, respectively; $a_1$ the fitting coefficient.

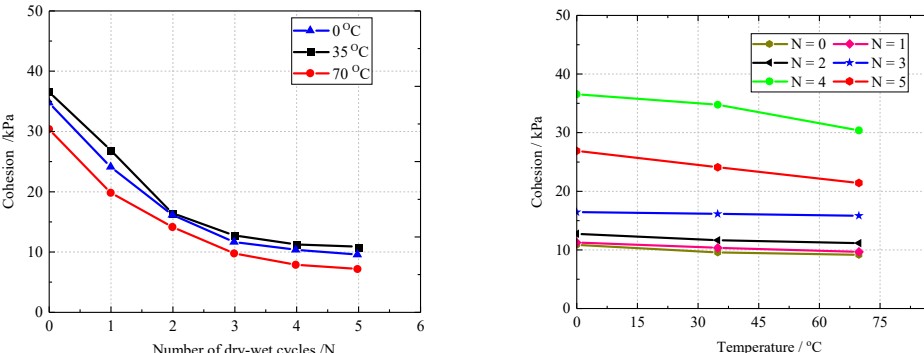

**Figure 7.** Cohesion curves of soil specimens under dry–wet cycling and temperature.

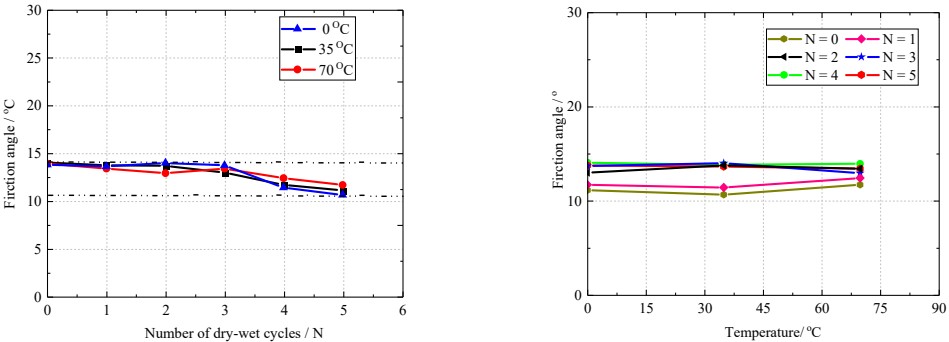

**Figure 8.** Variation in the friction angle values with dry–wet cycling and temperature.

The cohesion of unsaturated soil $c_1$ could be expressed as

$$c = c_T + (u_a - u_w) \tan \phi^b \tag{4}$$

where $c$ is the cohesion of unsaturated soil, $u_a$ is the pore air pressure, $u_w$ is the pore water pressure, $\phi^b$ is the friction angle.

Under the assumption that $\tan \phi^b$ was independent of the temperature [36], the C-M-S cohesion $c$ was be expressed as

$$c = c_0 + a_1(T - T_0) + (u_a - u_w) \tan \phi^b \tag{5}$$

Shear strength $\tau_{Tf}$ of the CMS subjected to a given temperature could be expressed as

$$\tau_{Tf} = c_0 + a_1(T - T_0) + (u_a - u_w) \tan \phi^b + (\sigma - u_a) \tan \phi' \tag{6}$$

where $(\sigma - u_a)$ denotes the normal net stress, $\tau_{Tf}$ denote shear strength for temperature $T$ of the CMS, $\phi'$ is the net friction angle.

From Figure 8, we noted that the friction angles for all specimens almost unchanged with the values fluctuating between 14.00° and 10.60°. This was probably because the grain composition of the CMS did not vary under the application of dry–wet cycling and temperature. The saturated sample was used in the triaxial test to make the grain gap smaller, leading to a small change in the friction angle.

Comparing Figures 7 and 8, the effect of dry–wet cycling on the cohesion was greater than that on the friction angle. The degradation of cohesion was more significant than the friction-angle variation. To further understand how the shear-strength deterioration of the CMS changed with dry–wet cycling and temperature, we next obtained the cohesion degra-

dation degree $D$. The parameter $D$ had been applied to measure changes in the material strength owing to environmental conditions [37], and it could be expressed as follows:

$$D = \left(1 - \frac{S_N}{S_O}\right) \times 100\% \tag{7}$$

where $D$ is the degradation degree of cohesion, $N$ indicate the number of cycles, $S_N$ is the C-M-S cohesion after dry–wet cycling, and $S_o$ is the C-M-S cohesion not subjected to dry–wet cycling.

From Table 2 and Equation (7), we could infer that the cohesion decreased with increasing in $N$. For cycles 1, 2, 3, 4, and 5, the average degradation degrees of cohesion were 29.04%, 52.47%, 65.34%, 69.61%, and 71.30%, respectively. The result suggested that the cohesion significantly degraded at first, and then gradually degraded with the increase of dry–wet number. It was obvious that after the third cycle, the particles of the CMS tended to be evenly arranged, and the surface friction tended to be stable.

To quantitatively study the degradation of cohesion during dry–wet cycling, an exponential-function-fitting analysis was conducted on the cohesion degradation degree. The fitting results are presented in Figure 9 and Table 3. The fitting function could be expressed as follows:

$$D(\mathrm{N}) = a - b \cdot \exp\left(-\frac{N}{d}\right) \tag{8}$$

where $a$ is the final degradation degree, $b$ is the parameter that controls the deterioration rate (a larger $b$ corresponds to faster cohesion degradation), and $d$ is the fitting parameter.

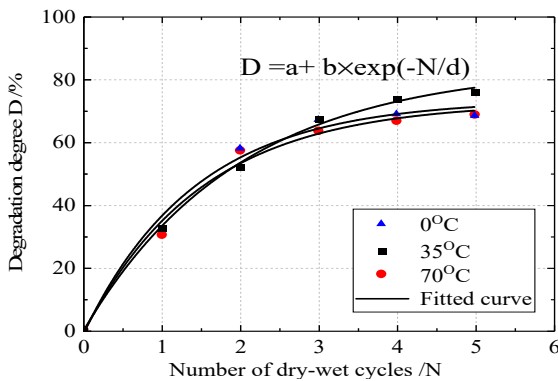

**Figure 9.** Degradation of cohesion under dry–wet cycling.

From Table 4, we noted that the correlation index $R^2$ fitted by Equation (8) were mostly greater than 0.9; thus, it was clear that Equation (8) suitably fitted the relationship between the deterioration degree and the $N$.

**Table 4.** Fitting parameters of cohesion degradation.

| Parameter | 0 °C | 35 °C | 70 °C |
|:---:|:---:|:---:|:---:|
| $a$ | 84.18 | 73.47 | 72.74 |
| $b$ | 84.45 | 74.60 | 73.80 |
| $d$ | 1.97 | 1.43 | 1.50 |
| $R^2$ | 0.99 | 0.98 | 0.98 |

Under the assumption that $\tan\phi^b$ was independent of $N$, we could rewrite cohesion $c_N$ as

$$c_N = \left[1 - a + b \cdot \exp\left(-\frac{N}{d}\right)\right]\left[c_0 + (u_{\mathrm{a}} - u_{\mathrm{w}})\tan\phi^b\right] \tag{9}$$

where $c_N$ and $c_0$ denote the cohesion for $N$ and initial cohesion, respectively $a_1$ the fitting coefficient; $(u_a - u_w)$ is the suction, which is obtained by the pressure plate device method.

The change in shear strength $\tau_{Nf}$ of the CMS with $N$ can be expressed as

$$\tau_{Nf} = \left[ 1 - a + b \cdot \exp\left( -\frac{N}{d} \right) \right] \left[ c_0 + (u_a - u_w) \tan \phi^b \right] + (\sigma - u_a) \tan \phi' \qquad (10)$$

where $(\sigma - u_a)$ denotes the normal net stress, $\tau_{Nf}$ denote C-MS shear strength with $N$, $\phi'$ is the net friction angle.

### 3.3. Quantitative Analysis of C-M-S Microstructure

The average variation in the microscopic parameters as a function of dry–wet cycle and temperature is shown in Figure 10. From Figure 10 we noted that the mean pore fractal dimension increased linearly and the mean probability entropy decreased exponentially with increasing in $N$. This result indicated that the reaction of water with CMS led to soil-defect expansion. The pore fractal dimension increased owing to the water-sensitivity of the minerals; the flaky microcrystallites separated the pores into many micropores, soil-particle aggregation decreased, and irregular collocation weakened. In general, microfractures reduced the C-M-S integrity, thereby leading to a greater degree of particle reorientation. The observed average change in these microstructural parameters proved that dry–wet cycling significantly affected the structure of the CMS. Meanwhile, from Figure 10, we noted that the mean fractal dimension and mean probability entropy remain nearly unchanged when the temperature was below 35 °C. Above 35 °C, the mean fractal dimension prominently increased with increasing temperature, corresponding to a significant increase of the pore number of particles under high temperature.

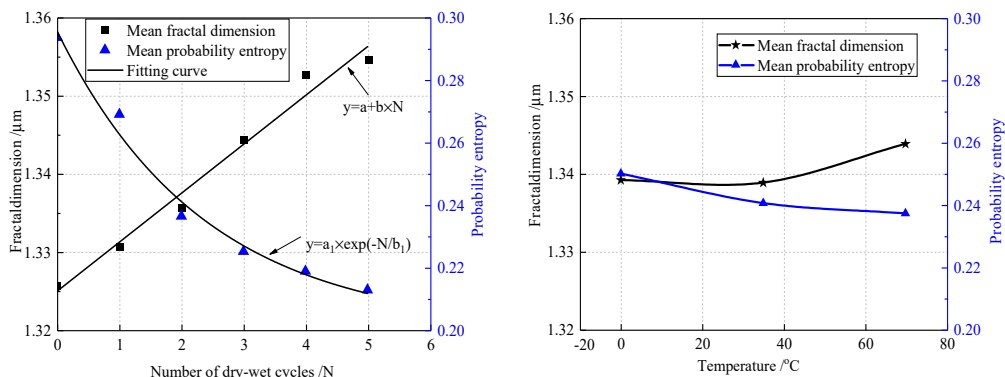

**Figure 10.** Variation in the microscopic parameters under dry–wet cycling and temperature.

Next, to explore the correlation between the microscopic and strength parameters of the CMS, we considered the two microscopic parameters (fractal dimension and probability entropy) and the cohesion degradation degree.

From Figure 11 we observed that the cohesion decreased with increase of the fractal dimension, with this reduction exhibiting a quadratic trend. The fractal dimension was characterized by the internal-structure porosity and arrangement, and the increase of fractal dimension corresponds to the increase of particle porosity and greater dispersion of soil particles. The cohesion of fine-particle soil was mainly owing to the liquid bridge force generated by water bound between soil particles and the combined "connecting" forces due to the cementation of capillary water and soil. The presence of bound water surrounding soil particles reduced the cohesive force of the fine-particle soil. Consequently, we could conclude that the liquid bridge force and connection force formed by the bound water were greatly reduced under the conditions of dry–wet cycling and high temperature.

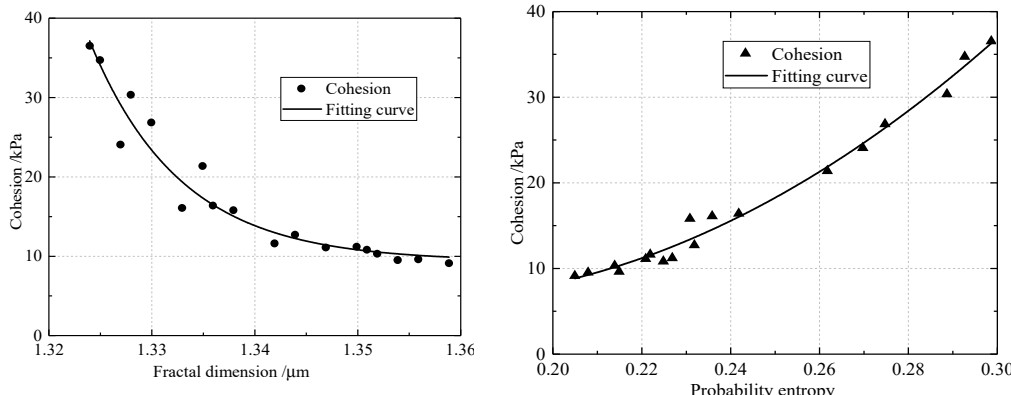

**Figure 11.** Variation in the C-M-S cohesion with the microscopic parameters.

In contrast, from Figure 11, we noted that the soil cohesion exhibited a positive correlation with the probability entropy; this result indicated the significant unidirectional influence of dry–wet cycling on the microstructure, which was reflected as a significantly weakened pore orientation degree, weakened dictional arrangement, and reduced soil probability entropy. The smaller probability entropy corresponded to a greater anisotropy rate, which resulted in a decrease of the shear strength.

The quantitative relationship between cohesion $c$ and the two microscopic parameters could be expressed as:

$$\left.\begin{array}{c} c = 3.948 \times exp(\frac{D_\rho}{0.009}) + 9.245 \\ R^2 = 0.935 \end{array}\right\} \tag{11}$$

$$\left.\begin{array}{c} c = 1722.530H_n{}^2 - 575.040H_n + 54.197 \\ R^2 = 0.9856 \end{array}\right\} \tag{12}$$

where $c$ denotes the cohesion, $D_\rho$ denote C-M-S fractal dimension, $H_n$ denote C-M-Sprobability entropy.

## 4. Discussion

### 4.1. Microstructure Variation on Dry–Wet Cycling

Dry–wet cycling reduces the amount of particle aggregation and particle contact, which can weaken the bonding between particles and fine pores. Moreover, dry–wet cycling changes the C-M-S microstructure from the stable surface-to-surface contact to the chain structure; this process is different from that reported by Han [29] as regards microstructure change. In the study, we examined the change in microstructure from surface-to-surface to surface-to-edge contact for different amounts of water. We found that dry–wet cycling increases the edge-debris amount, degree of particle fragmentation, and discontinuities. Notably, the C-M-S minerals are mainly composed of clay minerals; the montmorillonite/illite concentration was 12.5%, accounting for 20.83% of the clay mineral composition [17], thereby indicating that clay minerals are widely distributed in CMS and exist in various forms. Here, we noted that clay minerals exhibit a high swelling-shrinking ability under dry–wet cycling [38–40]. Next, water is a polar molecule and solvent, and CMS contains a variety of soluble components such as clay minerals [41]. These components may amplify the interactions between water and CMS. With the increase of $N$, the contact relationship gradually changes. The change of the contact relationship between the basic structural units of particles leads to significant softening deformation of CMS. The presence of water could lead to the generation of liquid bridge forces, which could modify the clay-mineral interparticle forces [42]. Under dry–wet cycling, the water molecules in clay minerals evaporated leading to soil shrinkage and structural change under dry environments, the liquid-bridge of water accelerated the structural damage under wet conditions [43]. After dry–wet cycling, the liquid-bridge and interparticle forces cluster the clay-mineral particles. Therefore, the C-M-S porosity is more after dry–wet

cycling. After a certain $N$, C-M-S mineral ions precipitate and the solution concentration increases until saturation [44].

### 4.2. Temperature-Induced Stress

The development of cracks in the CMS reflects the soil sample affected by temperature. From Figure 12, we observed six prominent cracks on the sample surface for the temperature setting of 0 °C. At 35 °C, cracks began to develop and penetrate the soil, thereby generating a small number of tiny cracks. The sample surface exhibited increased fragmentation due to fissures. At 70 °C, the primary cracks on the surface were connected, and the soil particles at the crack edges were dissolved. The cracks became blurred and more tiny cracks were formed. Here, we focused on the prominent cracks in the C-M-S samples at different temperatures. However, the smaller fissures gradually increased with increasing temperature. Thus, we could conclude that the temperature effected the soil pore structure mainly in the form of smaller-cracks development.

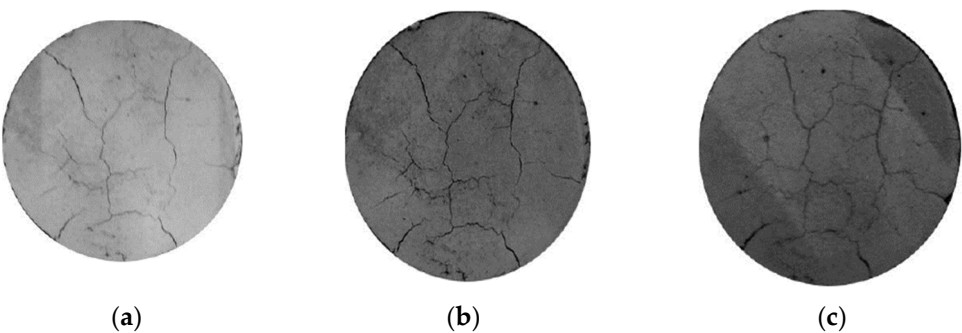

|        |        |        |
|:------:|:------:|:------:|
| (**a**) | (**b**) | (**c**) |

**Figure 12.** Crack generation in C-M-S samples at different temperatures after three dry–wet cycles: (**a**) T = 0 °C, (**b**) T = 35 °C, (**c**) T = 70 °C.

Next, to explore the reason for the temperature's cracking of soil samples, we analyzed the thermal stress of the C-M-S sample. As per thermodynamic principles, thermal expansion of the soil constrains the generated thermal stress. Soil expansion and stress are caused by changes in temperature. The temperature-induced stress could be obtained as follows [45]:

$$\sigma_{X(Y)} = \frac{\alpha E}{1-\mu}\left[-T(Z) + \frac{1}{L-0.01}\int_{0.01}^{L} T(Z)dZ\right] \tag{13}$$

where $\alpha$ denotes the thermal expansion coefficient of the CMS ($\alpha = 0.5 \times 10^{-6}/°C$), $E$ the Young's modulus ($E = 2 \times 10^6$ Pa), $\mu$ is the Poisson's ratio ($v = 0.25$), and $L$ is the radius of CMS sample, $T(Z)$ is the distribution of temperature along the $Z$-axis.

The distribution of temperature along the $Z$-axis could be obtained as follows [41]:

$$T(Z) = T_0\left[1 - \frac{Z^2}{m}\right] \tag{14}$$

where $T_0$ denotes the temperature at the surface andm the temperature-distribution coefficient.

When $Z = 0.01$ m, $T(Z) = 35$, and when $Z = 0.0\ 4$ m, $T(Z) = 70$. Substituting these boundary conditions into Equation (14), we could calculate the distribution of temperature along the $X$-axis as follows:

$$T(Z) = 32.67\left[1 + 714.28Z^2\right] \tag{15}$$

$$\sigma_{X(Y)} = \frac{4}{3}\left[47335.53Z^2 - 32.66\right] \tag{16}$$

From Equations (15) and (16), we could infer that the temperature-induced stress distribution along the X-direction. Figure 13 depicts the temperature-induced stress distribution along the X-direction, plotted as per Equation (16), wherein we noted that the temperature decreases with increase of the internal depth of the sample. The compressive stress at the inner center had a maximum value of 0.44 Mpa, whereas the temperature-induced tensile stress at the outer boundary had a maximum value of 0.57 Mpa. The zero-stress point was located at Z = 0.015 m, which corresponded to the transition point between compressive and tensile stress. It could be observed that when $Z \in (0.015, 0.04)$, if the stress is greater than the tensile strength, cracks occur and develop. The tensile stress on the surface of the sample was relatively large, which explained the surface-crack increase after each dry–wet cycle and high-temperature exposure.

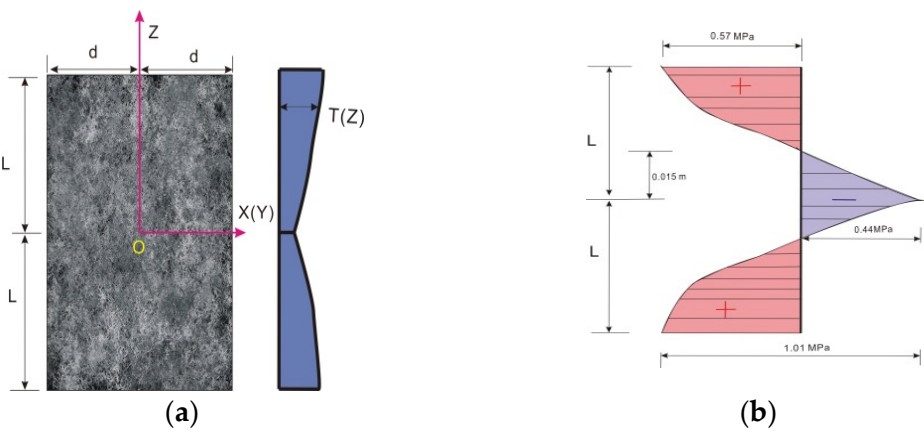

**Figure 13.** Temperature-induced stress distribution: (**a**) temperature distribution in the X–Z plane (**b**) temperature-induced stress distribution.

### 4.3. Failure Mechanism of the K209 Landslide

We next applied the above analysis to understand the failure mechanism of the K209 landslide. Strain softening, swelling–shrinkage, low strength, and high temperature of the CMS can be identified as internal factors underlying the K209 landslide. Because the C-M-S medium contained quartz, kaolinite, and other minerals, several mixed-layer clay minerals were formed. The crystal units of montmorillonite/illite were composed of a three-layered structure. Because of the weak van der Waals force corresponding to the interlayer connections, exchange cations and water molecules could easily enter the crystal cells. The presence of soluble salt ions increased the thickness of the diffuse double layer of clay particles. In particular, the presence of sodium ions in CMS significantly increased the diffusion-layer thickness. In addition, the distance between the cells increased with the thermal effect and the entry of water molecules, thus leading to soil expansion (Figure 14). When the repulsive forces were greater than the attractive forces between the clay platelets, the individual clay platelets separated from the surface of the soil and formed a suspension in water [46]. The microstructure of these minerals could be damaged during dry–wet cycling and exposed to high temperatures and rainfall; thus, the soil porosity gradually increases, resulting in reduced soil shear strength. Once the shear strength dropped to a critical value, the sliding of the soil body was accelerated, thereby leading to a landslide. To sum up, the combination of these peculiar features ultimately resulted in the deterioration of the engineering properties of landslide soils.

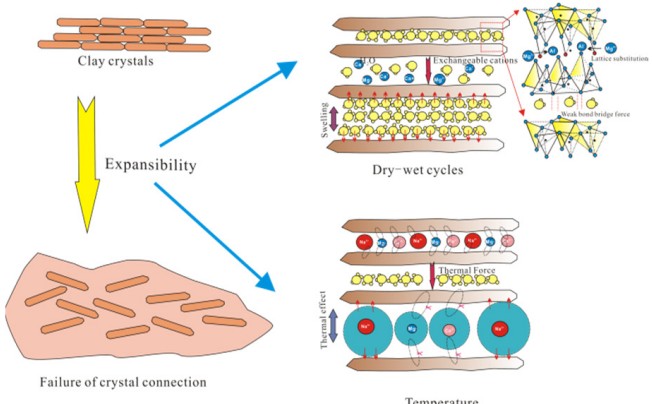

**Figure 14.** Failure mechanism of CMS subjected to dry–wet cycling and temperature.

## 5. Conclusions

In this study, we explored the effects of temperature and dry–wet cycling on the shear strength and microscopic parameters of C-M-S samples. Based on our results, we drew the following conclusions:

1. Dry–wet cycling and high temperatures significantly affect the shear strength. CMS exhibits obvious strain-softening properties. The soil cohesion is negatively correlated with the fractal dimension and positively correlated with the probability entropy.
2. The cohesion degeneration significantly increases before three dry–wet cycles; this degeneration can be satisfactorily described by an exponential equation function. Moreover, the cohesion exhibits a negative correlation with temperature. However, dry–wet cycling and temperature hardly influence the frictional angle.
3. Dry–wet cycling induces a significant change in the macroscopic properties of CMS. With the increase of $N$, the pore fractal dimension increases and the probability entropy decreases. The macroscopic mechanics of CMS are correlated with changes in the macrostructure parameters. Above 35 °C, temperature affects mainly the mean fractal dimension.
4. Temperature induces thermal tensile stresses on the sample surface. The surface-crack occurs once the high-temperature stress value is greater than 0.57 MPa.
5. Strain softening, swelling–shrinkage, low strength, and high temperature are the main factors affecting the engineering geology of C-M-S slopes; these factors form the material basis for rainfall-induced K209 shallow landslides. On the other hand, dry–wet cycling, temperature, and rainfall conditions are external factors that induce C-M-S-slope landslides.

**Author Contributions:** G.H.: Methodology, Conceptualization, Writing—-original draft, Investigation, Formal analysis, Visualization, Writing-review & editing. M.Z.: Supervision, Methodology, Funding acquisition. All authors contributed to this study. G.H. designed the research study and wrote this paper. M.Z. reviewed the manuscript. All authors have read and agreed to the published version of the manuscript.

**Funding:** This work was supported by the National Natural Science Foundation of China (NSFC) [Grant No. 51568022] and the Hubei Provincial Education Department Key Project [Grant No. B2021233].

**Institutional Review Board Statement:** Not applicable.

**Informed Consent Statement:** Not applicable.

**Data Availability Statement:** All data supporting the results of this study are included in the paper.

**Conflicts of Interest:** The authors state that they have no known competitive financial interests or personal relationships that could affect the work reported herein.

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
