# Peer review of "Effects of Dry–Wet Cycling and Temperature on Shear Strength and Microscopic Parameters of Coal Measure Soil"

_applsci, doi:10.3390/app13010336_

Round 1

Reviewer 1 Report

The manuscript is aiming at determining the shear strength parameters and mechanical behaviour of coal measure soil influenced by changing environmental conditions like temperature and changing saturation cycles. The manuscript is well-written and structured, although it lacks a detailed description of the methodology and approaches used in the study. The major concern is that the Authors completely neglected the unsaturated soil mechanics approach in testing. The authors use the formulas for expressing shear strength influenced by soil suction, but no information on how the suction was measured is provided.  The lab testing methods and procedures are missing a clear explanation. First of all how the wetting-drying process was performed (is it only a climate chamber, if so how the samples were kept, were they prepared the same way as for trx testing?). To understand the coupled phenomena of the behaviour the SWCC need to be provided and explained. The tested samples' composition needs to be explained. Not only physical but also chemical composition would be necessary to fully understand the results. How many samples were tested in total. What was the organic matter content and what was the content of soluble material (CaCO3, if any). These would defienietely influence the results . The authors mention salt ions being present, which could impact the suction.  The trx testing needs more explanation, to why CU test was applied (in this case it does not reflect in situ conditions). Was the suction measured at any stage of testing, or it is just the structure of the samples that would be considered? The values of the void ratio for each test (depending on the cycle) should be provided too. Please focus more on interpreting the SEM data. The authors provide the equation for the shear strength, where effective stress is confused with net stress (please see the marked copy of the manuscript). The chapter on Failure mechanism of the K209 landslide, seems to be very important for the entire research (since that was the aim, to understand it) but it is just briefly touched. More critical review in that matter is recommended The authors should limit the content to focus more on understanding mechanical behaviour. Instead of presenting results on cracking analyses extend the explanation on shear tests. The cracking analyses are just briefly provided using mechanical parameters, that are not discussed (how they are measured). These are only the major concerns, more specific comments and remarks can be found in the marked copy attached.

Author Response

Dear reviewer,

Thanks for your review of my article. Your comments and suggestions have been very helpful in improving my paper. I have made the following modifications to the paper according to your suggestions. May we kindly ask you to review it again and give us your valuable comments?

Best regards,

Gang Huang

Responses to Reviewer 1:

(1). Reviewer question :

Please explain what is k209 type landslide.

Responses to reviewer:

     k209 type landslide is a landslide studied in this paper and is numbered slope K209. According to the question, k209 type landslide has been revised to k209 landslide.

(2). Reviewer question:

 The aim of the study should be clearly stated in the abstarct

Responses to reviewer:

According to the question, We've increased the aim of the study: The aim of this study is to experimentally investigate the effects of dry–wet cycling and temperature on shear strength and microscopic parameters of coal measure soil. We carried out an unconsolidated undrained triaxial tests and scanning electron microscopy of CMS obtained from the K209 slope on the Chang-li highway

(3). Reviewer question :

How shallow landslides are defined her. What depth is consider, what are the threshold values. Please be more specific.

Responses to reviewer:

According to the question, We've increased the define of shallow landslides: (generally less than 2m deep), and the failure surface is parallel to the slope surface. 30%~50% of the landslides have reoccurred even after treatment.

(4). Reviewer question :

Wetting and drying cycles applications are strictly connected to changes of hydraulic properties of the soil. The intorduction should include comments on how suction influences the shear strength. IS the unsaturated soil mechanics approaches consider in the present study?

Responses to reviewer:

According to the question, We' ve increased suction study in the present study.

Maio[18] proved that the suction had a positive effect on the shear strength. Fredlund[19] matric suction affected the adsorption strength and friction Angle between soil particles, thus affecting the shear strength.

[18] C. Maio and G. Scaringi.. Shear displacements induced by decrease in pore solution concentration on a pre-existing slip surface. Eng. Geol. 2016, 200: 1–9.

[19] Fredlund, D. G., H. Rahardjo, and M. D. Fredlund. Unsaturated soil mechanics in engineering practice. Hoboken, NJ: Wiley, 2012.

(5). Reviewer question :

 Please increase the quality of the figures. The captions need to fit the figure, please correct

Responses to reviewer:

It has been changed as recommended.

(6). Reviewer question :

 Please provide the reference

Responses to reviewer:

It has been changed as recommended.

J Yang, M Zheng. Soil-water characteristic curve of coal measure soil under the influence of density and dry-wet cycle. Journal of East China Jiaotong University,2018,35(03):91-96.

(7). Reviewer question :

 Please provide the standard as a reference.

Responses to reviewer:

According to the question, We've increased the Chinese standard GBT50123-2019

The Professional Standards Compilation Group of People's Republic of China. GBT50123-2015. Design stand-ard of highway embankment [S]. Beijing: China Communications Press, 2019.

(8). Reviewer question :

 The procedure of wetting and drying needs to be explained in details

Responses to reviewer:

According to the question, We've increased the content: We injected 10% of water by weight on the upper and lower sides of the sample and stored water for 4h. Then, the samples were heating-equipment-dried at 70°C for 8h. The total period of each dry-wet cycle was 12 h.

(9). Reviewer question :

 Do you mean bulk density? Please use precise term

Responses to reviewer:

According to the question, we confirm that it is the bulk density, which has been modified.

(10). Reviewer question :

 Does it mean the wetting and drying was commenced in climate chamber only? In case of wetting and drying cycles the crucial information comes from SWCC. Show in the results of hydraulic behavior is required here.

Responses to reviewer:

The wetting and drying was commenced in climate chamber. SWCC has been added as recommended.

The hydraulic behavior of unsaturated soils can be characterized by the soil water characteristic curves (SWCCs) under wetting and drying cycles. We performed the axis translation technique ( the pressure plate device ) to obtain SWCC. The SWCC and the van Genuchten (1980) equation fit of the C-M-S specimens are shown in Figure3. It is noted that the air entry value of not cycle and 5 cycles is about 20 kPa and 11 kPa, respectively. The dry–wet cycling results in the densification of C-M-S unsaturated soils, which leads to changes in SWCC.

(11). Reviewer question :

 The physical parameters need to be listed in taht section. The quastion is whast is the cemicla composition. Is there any content of CaCO3 for instance, any organic matter? Please provide such information.

Responses to reviewer:

  • It has been added as recommended, see the revised draft.

As described on Table 2, chemical compositions of the C-M-S specimen were mainly SiO2, K2O, CaO and Al2O3.

Table 2. Chemical compositions of the C-M-S specimen.

Specimen

SiO2

K2O

CaO

Al2O3

F2O3

MgO

Na2O

Loss on ignition

CMS

32.54

17.24

15.20

14.58

2.24

1.40

0.75

16.05

 (12). Reviewer question :

 The procedure for wetting and drying using climate chamber needs to be explained in details, to veryfi boundary conditions, factors affecting the sample structure changes.

Responses to reviewer:

It has been added as recommended.

Polyvinyl chloride wrapped on the laterally sides. We injected 10% of water by weight on the upper and lower sides of the sample and stored water for 4h. Then, the samples wrapped by Polyvinyl chloride and rubber film were dried - at 70°C for 8h in climate-chamber.The total period of each dry-wet cycle was 12 h.

(13). Reviewer question :

 Why the authors decided to use CU methods, that does not reflect the real insitu conditions.

Responses to reviewer:

This triaxial test is unconsolidated consolidated undrained (UU) test. There was a mistake in writing, which has been corrected.

(14). Reviewer question :

 How many samples where tested in total?.

Responses to reviewer:

There are 54 triaxial test samples in total.

(15). Reviewer question :

 The crucial information that is missing is how was the soil suction measured or controlled. The authors refer to unsaturated soil mechanics showing the formula based on values for air pore pressure but no such data is provided.

Responses to reviewer:

We performed the pressure plate device method (the axis translation technique) to measure suction. The figure shows the change of matric suction

(16). Reviewer question :

 To assume the "changed" int. friction angle is temp. dependent the reference needs to be provided.

Responses to reviewer:

It has been added as recommended.

[1] Yavari N ,  Tang A M ,  Pereira J M , et al. Effect of temperature on the shear strength of soils and the soil–structure interface[J]. Canadian Geotechnical Journal, 2016, 53(7).

(17). Reviewer question :

 Please make sure all the symbols are explained

Responses to reviewer:

  • It has been added as recommended, see the revised draft.

 (18). Reviewer question :

 This is effective stress. net stress includes por air pressure

Responses to reviewer:

  • It has been added as recommended, see the revised draft.

(19). Reviewer question :

 How suction value determined, how the suction values were obtained?

Responses to reviewer:

We performed the pressure plate device method (the axis translation technique) to obtain suction.

(20). Reviewer question :

 This also influances the suction, what about the soluable material content that influances the structure after the wetting cycles. Was that taken into account?

Responses to reviewer:

The moisture content of strength test study was constant in this manuscript, and the influence of suction on the strength was relatively small, which was not considered in the analysis of the results. The influence of soluble substance content on the structure after drying and wetting cycle has not been considered due to the limited manuscript space. In the future, we will focus on studying the change of suction of coal measure soil and considering the influence of soluble substance content on the structure. Thank the reviewer for their advice.

 (21). Reviewer question :

 More critical review of the approach should be provide, there are number of simplifications applied in the lab testing apporach (unsatirated soil mechanics criteria are mostly neglected during testing).

Responses to reviewer:

We have added SWCC and suction parts of unsaturated soil in this paper. Due to limited space of this paper, there are no quantitative studies on the effects of dry and wet cycles on suction and soluable material content. In view of the importance of suction to strength, in the follow-up to strengthen this aspect of research. Thank the reviewer for their advice.

Reviewer 2 Report

Applied Sciences

Manuscript ID: applsci-1932471

Effects of dry–wet cycling and temperature on mechanical and microscopic characteristics of coal measure soil

by

Gang, Mingxin

REFEREE’S COMMENTS

The study is on a subject that should be of great interest to many readers. For location of comments, please see the belows.

  1. Title:
  2. It should be more focused/concise.
  3. Abstract:
  4. The abstract should be supported by quantitative findings.
  5. Keywords:
  6. Remove the "1. Introduction" from the list of keywords.
  7. Introduction:
  8. See the study in Periodica Polytechnica Civil Engineering 60 (4), 603-609; and other similar papers already available in the literature.
  9. Last paragraph: The authors should clearly indicate the originality/novelty of their research. A separate paragraph (the last paragraph) would be better for this purpose.
  10. Materials and Methods:
  11. Figure 1; a-b: the coordinates are not seen clearly.
  12. Keep the maps in their original size to avoid any misunderstanding.
  13. Figure 2; no need to use different colors. Black-white would be recommended.
  14. Triaxial tests: how did the authors measure the strain and strength properties?
  15. Figure 4.a is not necessarily required.
  16. Describe the Figure 5 with more details in the text.
  17. How did the authors control the temperature in the laboratory?
  18. Test Result Analysis:
  19. The authors should use same formatting in the plot areas. For example, light-gray-dotted-gridlines would be very useful to follow the changes closely.
  20. Table 2; abbreviations instead of using full name would be better.
  21. line 254; "The parameter D has been applied ..."
  22. Equations 9-12 should be explained with more details. How did the authors derive them?
  23. Figure 10; the caption could be more concise/compact.
  24. Discussion:
  25. Figure 12; What is the reason of difference(s) in colour among the samples?
  26. What is the effect of water presence?
  27. Conclusions:
  28. The authors should extend the conclusions by referring the quantitative findings.
  29. In General:
  30. The language could be polished.
  31. Literature review should be extended.
  32. Check out the details of the references cited.
  33. Figures: light-gray-dotted gridlines in the plot areas would be useful for the potential readers in order to follow the changes.

Best regards,

Author Response

Dear reviewer,

Thanks for your review of my article. Your comments and suggestions have been very helpful in improving my paper. I have made the following modifications to the paper according to your suggestions. May we kindly ask you to review it again and give us your valuable comments?

Best regards,

Gang Huang

Responses to Reviewer 2:

(1). Reviewer question :

Title: It should be more focused/concise.

Responses to reviewer:

      According to the question, the title has been has been changed as recommended.

“Effects of dry–wet cycling and temperature on shear strength and microscopic parameters of coal measure soil”

(2). Reviewer question :

  Abstract: The abstract should be supported by quantitative findings.

Responses to reviewer:

It has been changed as recommended.

 The soil shear strength and microstructure parameters significantly decrease before three dry–wet cycles

The surface-crack occurs once the stress of high-temperature value is greater than 0.57MPa.

(3). Reviewer question :

Keywords: Remove the "1. Introduction" from the list of keywords..

Responses to reviewer:

It has been deleted.

 (4). Reviewer question :

 Introduction:

(a)See the study in Periodica Polytechnica Civil Engineering 60 (4), 603-609; and other similar papers already available in the literature.

(b)Last paragraph: The authors should clearly indicate the originality/novelty of their research. A separate paragraph (the last paragraph) would be better for this purpose.

Responses to reviewer:

(a)According to the question, We've increased related references :

[22] A. Cabalar and C. Clayton Effect of temperature on triaxial behavior of a sand with disaccharide. Periodica Polytechnica Civil Engineering, 2016, 60(4): 603-609.

(b) According to the question, It has been changed as recommended:

This paper firstly performed surveys on shear strength and microscopic parameters of coal measure soil under dry-wet condition and seasonal climate in the mountains of Southern China. In order to further explore the effects of dry–wet cycling and temperature on shear strength and microscopic parameters of coal measure soil, unconsolidated undrained triaxial tests and scanning electron microscopy investigations were carried out. Firstly, the effects of dry–wet cycling and temperature on the shear strength of C-M-S samples were investigated. Then, the microscopic parameter changes in the CMS after dry–wet cycling was studied .Finally, the crack development relationship with the tem-perature-induced stress after dry–wet cycling and the failure mechanism of the K209 landslide on the Chang-li highway were discussed.

(5). Reviewer question :

 Materials and Methods

(a)Figure 1; a-b: the coordinates are not seen clearly.

(b)Keep the maps in their original size to avoid any misunderstanding.

(c) Figure 2; no need to use different colors. Black-white would be recommended.

(d)Triaxial tests: how did the authors measure the strain and strength properties?

(e)Figure 4.a is not necessarily required.

(f)Describe the Figure 5 with more details in the text.

(g)How did the authors control the temperature in the laboratory?

Responses to reviewer:

(a) (b) (c) It has been changed as recommended.

(d) the axial strain rate is controlled by 0.05 mm/min, axial pressure was applied to the upper and lower surfaces of the soil specimen, The shear strength was considered as corresponding to the peak deviator stress or the deviator stress corresponding to 15% axial strain.

(e)Figure 4(a) is deleted.

(f)  It has been added as recommended: The plane fractal dimension of soil particle is a geometric object that indicates the aggregation of a soil in the plane space. The value of plane fractal dimension is obtained by the ratio of the cubic box length occupied by the total grids number occupied by the target as per the following formula.

 (g) The heating sheet was used to control the heating water, and cooling sheet was used to control the cooling water during the test.

(a )Geographical location      (b) Research area

(c) Landslide locations

Figure 1. Location of the study area and landslide grouping.

Figure 2. Grain-size distribution of the soil samples considered in the study.

 (6). Reviewer question :

 Test Result Analysis:

(a)The authors should use same formatting in the plot areas. For example, light-gray-dotted-gridlines would be very useful to follow the changes closely.

(b)Table 2; abbreviations instead of using full name would be better.

(c)line 254; "The parameter D has been applied ..."

(d)Equations 9-12 should be explained with more details. How did the authors derive them?

(e)Figure 10; the caption could be more concise/compact.

Responses to reviewer:

It has been changed as recommended.

(a) light-gray-dotted-gridlines are added to the figure,

(b) Table 2; abbreviations has been instead of using full name, see the revised draft.

(c) It has been revised to the parameter D

(d) )Equations 9-12 has been explained with more details, see the revised draft.

(e) Figure 10; the caption could be more concise/compact.

 (7). Reviewer question :

 Discussion

(a) Figure 12; What is the reason of difference(s) in colour among the samples? (b)What is the effect of water presence?

Responses to reviewer:

The black color is mainly due to increased soluable cations content after wetting and drying cycles and after high temperature

(8). Reviewer question :

 Conclusions: the authors should extend the conclusions by referring the quantitative findings.

Responses to reviewer:

It has been added as recommended, see the revised draft.

(9). Reviewer question :

 In General:

(a)The language could be polished.

(b)Literature review should be extended.

(c)Check out the details of the references cited.

(d)Figures: light-gray-dotted gridlines in the plot areas would be useful for the potential readers in order to follow the changes..

Responses to reviewer:

  • According to the question, the English throughout has been polished, see the revised draft.
  • (b)It has been extended as recommended, see the revised draft.
  • (c)According to the question, the references cited has been check outed, see the revised draft.
  • (d) It has been added as recommended, see the revised draft.

Round 2

Reviewer 2 Report

-